## [Reviewer comments · BMJ Open]

ARTICLE DETAILS

TITLE (PROVISIONAL)	Understanding the mechanisms of a combined physical and psychological intervention for people with neurogenic claudication: Protocol for a causal mediation analysis of the BOOST Trial
AUTHORS	Comer, Christine; Lee, Hopin; Williamson, Esther; Lamb, Sarah

VERSION 1 – REVIEW

REVIEWER	Masakazu Minetama Spine Care Center, Wakayama Medical University Kihoku Hospital, Japan.
REVIEW RETURNED	03-Feb-2020

GENERAL COMMENTS	Thank you for the opportunity to review this manuscript about study protocol for a causal mediation analysis of a RCT of physiotherapist-delivered, combined physical and psychological intervention. Although several researchers reported the efficacy of physical therapy for patients with LSS, it is still unknown how improvements affect patients-reported outcomes and walking capacity. As far as I know, no previous RCTs investigated the efficacy of combined physical and psychological intervention as conservative treatment for patients with LSS using detailed psychological assessments. I think that understanding of these treatment mechanisms is very important to generate evidence that can be used to tailor and refine treatments and optimize clinical effectiveness. It was unfortunate that MRI scan was not used to diagnose LSS. In the present study, a validated LSS diagnostic support tool was used for the diagnosis of LSS. Pain of the lower extremities due to osteoarthritis should affect the score. In this study, OA of the lower extremity was not involved in the execution criteria. Overall, I am very impressed with the study. However, before its publication, there are a few points where they can improve this manuscript. The description of Figure 1 does not appear in the text. On p. 5, 9-12, reference 34 does not seem to relate to this sentence.
---

REVIEWER	Carlo Ammendolia University of Toronto, Toronto, Canada
REVIEW RETURNED	08-Feb-2020

GENERAL COMMENTS	Thank you for the opportunity to review this protocol entitled "Understanding the mechanisms of a combined physical and psychological intervention for people with neurogenic
---

	claudication causes changes in pain and disability: Protocol for a causal mediation analysis of the BOOST Trial". Below are my comments. The protocol is very well written except for the title which I find confusing. Perhaps shortening the title to: "Understanding the mechanisms of a combined physical and psychological intervention for people with neurogenic claudication: Protocol for a causal mediation analysis of the BOOST Trial" The background information is concise and provides a sound rationale for the study objectives and the use of causal mediated analysis. I would suggest clarification of the BOOST intervention description on Page 7 line 23 regarding the exercise program. Clarify that 60 minutes of exercise is performed at each session. Lines 24 and 26 on page 7 requires formatting. The selected potential mediators to be assessed are plausible and the authors use valid and reliable measures of the potential mediators. I would suggest considering depression and catastrophizing as potential mediators. A plausible comprehensive theoretical mediation framework is provided and well outlined in the figures. Another factor in the pathway that should be considered is treatment compliance. Although the statistical methods appear sound, I am not an expert in causal mediated analysis and suggest seeking an opinion from an expert in this area. The expert can speak to the potential strengths and weaknesses and underlying assumptions of the statistical methods. However, a power analysis is recommended to demonstrate the study's ability to adequately detect mediation. Only minor corrections of the manuscript are recommended. Thank you again for the opportunity to review this manuscript.
--	---

REVIEWER	Dana Goin University of California, San Francisco, USA
REVIEW RETURNED	17-Mar-2020

GENERAL COMMENTS	This an interesting application of causal mediation methodology to understand the mechanisms underlying potential results of trial of a physical and psychological intervention on disability among older adults. I have several suggestions which I think could improve the approach.  1. Clarify the type of mediation parameters that are of interest. The authors state that they plan to estimate direct and indirect effects that sum to the total effect; therefore, I assume they plan to estimate natural direct and indirect effects. This should be clarified as the assumptions necessary for identification of these effects differ from other types of direct and indirect effects. 2. Examine the correlations between all mediators of interest and examine them jointly. The authors state that they will only examine multiple mediator models if the secondary mediators are associated with the primary mediator. However, many of the secondary mediators may be correlated with one another, in addition to the primary mediator. Therefore, the authors should
--

	examine correlations between all potential mediators. Furthermore, the authors should consider examining all of the mediators jointly in addition to the individual mediation analyses they are planning. This will allow the authors to evaluate the proportion mediated by a set of mediators that may simultaneously be affected by the intervention, and reduce the likelihood there are exposure-mediator confounders that are affected by the intervention. 3. This most substantial challenge to the potential interpretation of results from the proposed analyses is the potential for exposure-mediator confounders that are affected by the intervention. In particular, natural direct and indirect effects are not identified when these types of confounders are present, even if measured. However, any such confounder is essentially another mediator, and can be included in multiple mediator analyses. For additional information, see VanderWeele T, Vansteelandt S. Mediation analysis with multiple mediators. Epidemiologic methods. 2014 Jan;2(1):95-115. 4. I am not a subject-matter expert, but I wonder whether several of the mediator-outcome confounders may be functioning as mediators when they are measured at the 6-month time point. For example, could general pain be affected by the intervention? 5. Another area in which clarification would be helpful is the description of “simulating” potential values for the mediators and outcomes. Are they describing a non-parametric bootstrap, or something else? For the bootstrapped confidence intervals, will they be using Wald-type confidence intervals, or percentile-based? 6. The authors state they will impute missing data only if the proportion of missing mediator or outcome data is greater than 15%. This is a much higher percentage than the standard 5%, and I recommend the authors reconsider their cutoff, or else give more context as to why they chose such a high percentage. Is there a possibility for missing covariate data, or do the authors know they will not have any missingness for these variables? If there is missingness in the covariates, will these also require imputation? Furthermore, can the authors add how many imputed data sets will they create, if they do impute?
--	---

VERSION 1 – AUTHOR RESPONSE

Authors' Responses to Reviewers comments	
Reviewer 1	
It was unfortunate that MRI scan was not used to diagnose LSS. Pain of the lower extremities due to osteoarthritis should affect the score.	Our trial is designed to reflect current practice in a primary/community care setting, where MRI scans are often not obtained or relied upon for a clinical diagnosis of LSS prior to treatment. We agree that diagnosis of LSS may be masqueraded by symptoms from lower limb conditions such as osteoarthritis, but believe that this is true with or without MRI investigation (that is known to have poor correlation with symptom presentation (Jensen et al 2020, Prevalence of lumbar spinal stenosis in general

	and clinical populations: a systematic review and meta-analysis. European Spine Journal, 1-21). We are also carrying out an MRI study as part of this work and will explore the relationship between MRI parameters, symptoms and outcomes.
The description of Figure 1 does not appear in the text.	Thank you for noting this omission – reference to Figure 1 is now included on Page 7 under heading Causal Mediation analysis-models
On p. 5, 9-12, reference 34 does not seem to relate to this sentence	Thank you for noting this reference error, which has now been corrected.
Reviewer 2	
The protocol is very well written except for the title which I find confusing. Perhaps shortening the title to: "Understanding the mechanisms of a combined physical and psychological intervention for people with neurogenic claudication: Protocol for a causal mediation analysis of the BOOST Trial"	Thank you for this suggested amendment of the protocol title. We have now amended the title accordingly.
I would suggest clarification of the BOOST intervention description on Page 7 line 23 regarding the exercise program. Clarify that 60 minutes of exercise is performed at each session.	We have expanded the description of the BOOST intervention following your recommendation, and agree that the more detailed description will improve appreciation of the context of the mediation analysis for readers.
Lines 24 and 26 on page 7 requires formatting	Formatting errors now corrected – thank you
I would suggest considering depression and catastrophizing as potential mediators.	In the BOOST trial, we have not measured catastrophizing, but have measured the related construct of fear avoidance beliefs and have included this as a potential mediator in our planned analyses. Whilst we appreciate the reviewer's suggestion to consider depression as a potential mediator, we have not included any formal measure of depression in the trial, as it is not a target for the BOOST intervention.
Another factor in the pathway that should be considered is treatment compliance.	A Complier Average Causal Effect (CACE) analysis has been conducted to assess the treatment effect when accounting for participants' compliance with the intervention and will be reported separately.

A power analysis is recommended to demonstrate the study's ability to adequately detect mediation.	Kenny and Judd (2014) have shown that in most settings, the power to detect indirect effects is greater than the power to detect total effects. The BOOST trial was powered to detect a between-group difference of 5 points on the ODI at 80% power, and 5% two-sided significance levels. A 5 point change is considered to be clinically significant, and baseline SD of 15 has been assumed, based on published estimates in older populations and those with NC. Therefore, we assume that the BOOST trial is adequately powered to detect indirect effects.
Reviewer 3	
1. Clarify the type of mediation parameters that are of interest. The authors state that they plan to estimate direct and indirect effects that sum to the total effect; therefore, I assume they plan to estimate natural direct and indirect effects. This should be clarified as the assumptions necessary for identification of these effects differ from other types of direct and indirect effects.	Thank you for requesting this clarification. We are interested in estimating natural indirect and direct effects that sum to the total effect. We have now outlined this in the statistical analysis section.
2. Examine the correlations between all mediators of interest and examine them jointly. The authors state that they will only examine multiple mediator models if the secondary mediators are associated with the primary mediator. However, many of the secondary mediators may be correlated with one another, in addition to the primary mediator. Therefore, the authors should examine correlations between all potential mediators. Furthermore, the authors should consider examining all of the mediators jointly in addition to the individual mediation analyses they are planning. This will allow the authors to evaluate the proportion mediated by a set of mediators that may simultaneously be affected by the intervention, and reduce the likelihood there are exposure-mediator confounders that are affected by the intervention.	Thank you for this suggestion. We agree that post-randomisation confounding of the mediator-outcome effect is problematic for the identification of natural indirect effects. To address this problem, we will include a joint mediator model that examines all six of our mediators simultaneously. Because we cannot be certain about the sequential ordering of our mediators, we will not be able to further decompose the effect into multiple path-specific effects using a joint mediator model. For this we will focus on our approach of refining the indirect effect through the primary mediator through each secondary mediator considered independently. Added text: "We will use a model-based inference approach for causal mediation"

	analysis [73]. All analyses will be conducted in R (The R Foundation for Statistical Computing) using the “medflex” and “mediation” packages [74]. To estimate a joint indirect effect, we will consider a natural indirect effect of the intervention on the outcome that is exerted through a vector of all six mediators (walking capacity, fear avoidance behaviour, walking self-efficacy, physical function, physical activity, and symptom severity). The advantage of this approach is that we can consider all possible mechanisms simultaneously and relax the assumption of omitting confounders of the mediator-outcome effect that is affected by randomisation. We will use the imputation-based approach outlined by Steen et al. (2017) to fit a natural effect model with robust standard errors based on the sandwich-estimator - the recommended approach when the ordering of multiple mediators is unknown.”
3. This most substantial challenge to the potential interpretation of results from the proposed analyses is the potential for exposure-mediator confounders that are affected by the intervention. In particular, natural direct and indirect effects are not identified when these types of confounders are present, even if measured. However, any such confounder is essentially another mediator, and can be included in multiple mediator analyses. For additional information, see VanderWeele T, Vansteelandt S. Mediation analysis with multiple mediators. Epidemiologic methods. 2014 Jan;2(1):95-115.	Addressed above
4. I am not a subject-matter expert, but I wonder whether several of the mediator-outcome confounders may be functioning as mediators when they are measured at the 6-month time point. For example, could general pain be affected by the intervention?	Addressed above.

5. Another area in which clarification would be helpful is the description of “simulating” potential values for the mediators and outcomes. Are they describing a non-parametric bootstrap, or something else? For the bootstrapped confidence intervals, will they be using Wald-type confidence intervals, or percentile-based?	We will be using the non-parametric bootstrap confidence intervals with the percentile method. Outlined by: Tingley D, Yamamoto T, Hirose K, Keele L, Imai K. mediation: R Package for Causal Mediation Analysis. J Stat Softw 2014;59:1–38. https://doi.org/10.18637/jss.v059.i05.
6. The authors state they will impute missing data only if the proportion of missing mediator or outcome data is greater than 15%. This is a much higher percentage than the standard 5%, and I recommend the authors reconsider their cutoff, or else give more context as to why they chose such a high percentage. Is there a possibility for missing covariate data, or do the authors know they will not have any missingness for these variables? If there is missingness in the covariates, will these also require imputation? Furthermore, can the authors add how many imputed data sets will they create, if they do impute?	We have revised our cut-off to a more conservative 5%. We do not expect more than 5% missing data for covariates. Revised text: “We will assess the proportion of missing mediator and outcome data. We will conduct all analyses on complete cases if the proportion of missing data is <5%. If missing data exceeds 5% we will use multiple imputation by chain equations to impute 10 datasets using the “mice” package [78]”

VERSION 2 – REVIEW

REVIEWER	Dana Goin University of California, San Francisco, USA
REVIEW RETURNED	17-Apr-2020
GENERAL COMMENTS	No additional comments.